# Projected rapid response of stratospheric temperature to stringent climate mitigation

Grasiele Romanzini-Bezerra [1] & Amanda C. Maycock [1] ✉

Deep, rapid and sustained reductions in greenhouse gas emissions are required to meet the 2015 Paris Agreement climate target. If the world strengthens efforts toward near-term decarbonisation and undertakes major societal transformation, this will be met with requests from policymakers and the public for evidence that our actions are working and there are demonstrable effects on the climate system. Global surface temperature exhibits large internal variability on interannual to decadal timescales, meaning a reduction in the magnitude of surface warming would not be robustly attributable to climate mitigation for some time. In contrast, global stratospheric temperature trends have much higher signal-to-noise ratios and could offer an early indication of the effects of climate mitigation. Here we examine projected near-term global temperature trends at the surface and in the stratosphere using large ensemble climate models following three future emission scenarios. Under rapid, deep emission cuts following SSP1–1.9, modelled middle and upper stratospheric cooling trends show a detectable weakening within 5 years compared to a scenario approximately representing current climate commitments (SSP2–4.5). Therefore, stratospheric temperature trends could serve as an early indicator to policymakers and the public that climate mitigation is taking effect.

The United Nations Environment Programme 2023 Emissions Gap Report[1] concluded the world is not currently on track to meet the 1.5 °C global surface temperature target of the UNFCCC 2015 Paris Agreement. The remaining carbon budget to limit warming to 1.5 °C is rapidly diminishing[2], and the world remains focused on developing effective climate policies that keep 1.5 °C within reach. If society implements stringent reductions in greenhouse gas emissions in line with the Paris temperature target, a natural question that will follow is "are our actions working"? Detecting the signal of mitigation in surface climate variables like global surface air temperature (GSAT;[3–5]) or Arctic sea ice is complicated by the presence of large internal climate variability (Fig. 4.15 in ref. [6]), which confounds the detection of externally forced trends, as well as by the lagged response of many aspects of the climate system to changing atmospheric greenhouse gas concentrations due to ocean inertia[7]. We may, therefore, need to

look elsewhere to see the first signs that the climate system is being steered onto a different track.

The canonical pattern of atmospheric temperature change due to increased greenhouse gas concentrations comprises global average tropospheric warming and stratospheric cooling (e.g.,[8]). This vertical 'fingerprint' is a key indicator of human influence on the climate[9]. On timescales longer than a month or so, the globally-averaged stratosphere is close to radiative equilibrium, with a balance of shortwave heating and longwave cooling primarily due to the presence of radiatively active gases ozone, $CO_2$, and water vapour[10]. The fact that in the global average, the stratosphere is close to radiative balance means global stratospheric temperatures exhibit much lower internal variability than in the troposphere, where strong coupling with the global oceans drives substantial interannual to decadal variability[3,7,11]. This means externally forced temperature trends in the stratosphere are

[1]School of Earth and Environment, University of Leeds, Woodhouse Lane, Leeds LS2 9JT, UK. ✉e-mail: a.c.maycock@leeds.ac.uk

more readily detectable. Indeed, it was noted in the IPCC First Assessment Report (1990)[12] that 'stratospheric cooling alone has been suggested as an important detection variable'. Santer et al.[9] quantified the anthropogenic signal in historical atmospheric temperature trends and showed that including middle and upper stratospheric data alongside tropospheric layers improves the detectability of the anthropogenic fingerprint by a factor of five.

Observed stratospheric cooling over the past 40 years has been primarily attributed to increasing greenhouse gases and stratospheric ozone depletion[13–17], with a smaller role for stratospheric water vapour changes[18]. In the future, projected stratospheric cooling driven by rising greenhouse gases will be partly offset by height-dependent warming from stratospheric ozone recovery[19]. Nevertheless, we still anticipate higher signal-to-noise ratios in the stratosphere, which offers a potential route to more rapid detection of the effects of mitigation on climate.

As a result of internal variability, there are many plausible trajectories the climate system could take under the same future boundary conditions and external forcing[20]. Here, we use large initial condition ensemble projections from three climate models, which enables the externally forced climate change signal (represented by the ensemble mean) to be assessed relative to the amplitude of internal variability (represented by the ensemble spread). Within the large ensemble framework, one can think of the observed climate trajectory as roughly analogous to a single member drawn from the scenario that most closely matches our future emissions path. Based on the model projections, we show that satellite-observable stratospheric temperature trends could provide a clear sign that stringent mitigation is altering the climate trajectory within ~5 years.

## Results

We examine four global average temperature indicators (see Methods): GSAT and three well-established satellite-observable atmospheric layers covering the temperature lower stratosphere (TLS); temperature middle stratosphere (TMS); and temperature upper

stratosphere (TUS). These indicators are expected to be observable in the near future using existing and planned measurement platforms (e.g., ref. [21]).

We focus on comparing the temperature indicators in three future emissions scenarios from the CMIP6 ScenarioMIP project[22]: SSP1–1.9, SSP2–4.5, and SSP3–7.0. These scenarios serve as indicative pathways that broadly represent the successful implementation of stringent near-term mitigation towards the Paris Agreement 1.5 °C target (SSP1–1.9), successful implementation of current mitigation policies and commitments (SSP2–4.5), and failure to implement current commitments (SSP3–7.0) (see Methods).

Figure 1 shows projections of the global temperature indicators for the CanESM5 model over the period 2023–2045. The TMS and TUS timeseries show stratospheric cooling superposed with externally forced decadal variability associated with the solar cycle. Notably, for TMS and TUS, the ensemble spread is small compared to the ensemble mean changes, confirming the high signal-to-noise level (see also ref. [9]). As expected, GSAT shows a monotonic increase in all scenarios consistent with the assessment of ref. [6], but the ensemble spread relative to the magnitude of the ensemble mean anomaly is considerably larger than for TMS and TUS. TLS shows relatively weak ensemble mean anomalies in the near term and proportionately larger ensemble spread.

Figure 2 shows near-term global temperature trends for the three emissions scenarios over the next 5, 10, 15, and 20 years (see Methods). The three climate models show different rates of future GSAT warming, in large part because they have different transient climate responses (TCR) and effective climate sensitivities (ECS)[23] (see Methods). In all three models, the spread of GSAT trends amongst ensemble members is strongly overlapping between the emissions scenarios for at least the next 10–15 years. Therefore, we could not expect to confidently detect and attribute any effect of climate mitigation on the rate of surface warming in the near future (cf. in ref. [3]). In the lower stratosphere (TLS), the projections show a weak cooling trend in the next 5 years, which progressively reduces in amplitude over a 15–20

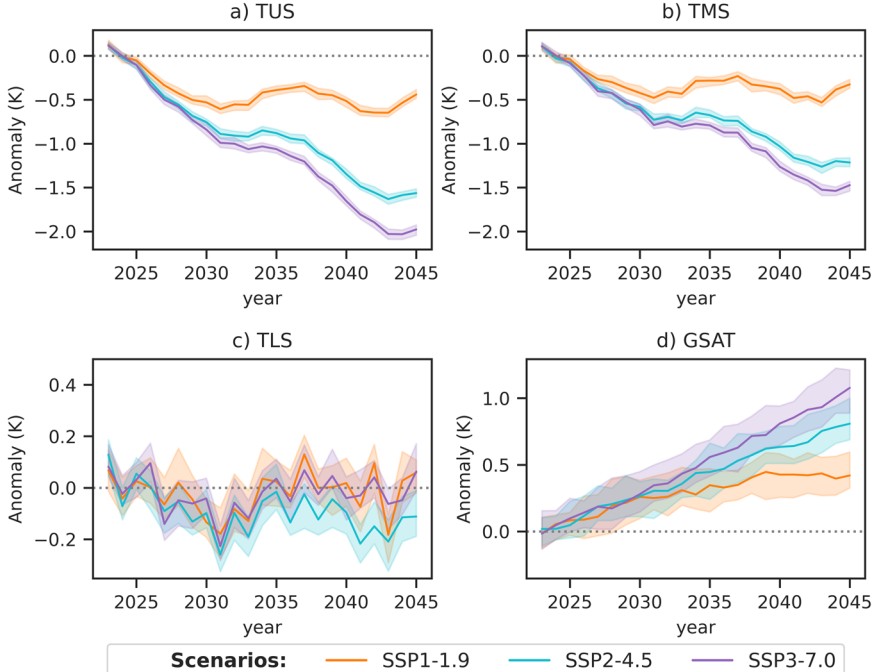

**Fig. 1 | Timeseries of global annual mean temperature anomalies in CanESM5.** Timeseries covering the period 2023–2045 for **a** temperature of the upper stratosphere (TUS), **b** temperature of the middle stratosphere (TMS), **c** temperature of the lower stratosphere (TLS), and **d** global surface air temperature (GSAT). Anomalies are defined relative to the period 2021–2025. Colours show the three future emissions scenarios. Thick lines show the ensemble mean, and shading denotes the 10–90th percentile range of ensemble members. Note the different *y* axis ranges in the subpanels. Source data are provided as a Source Data file.

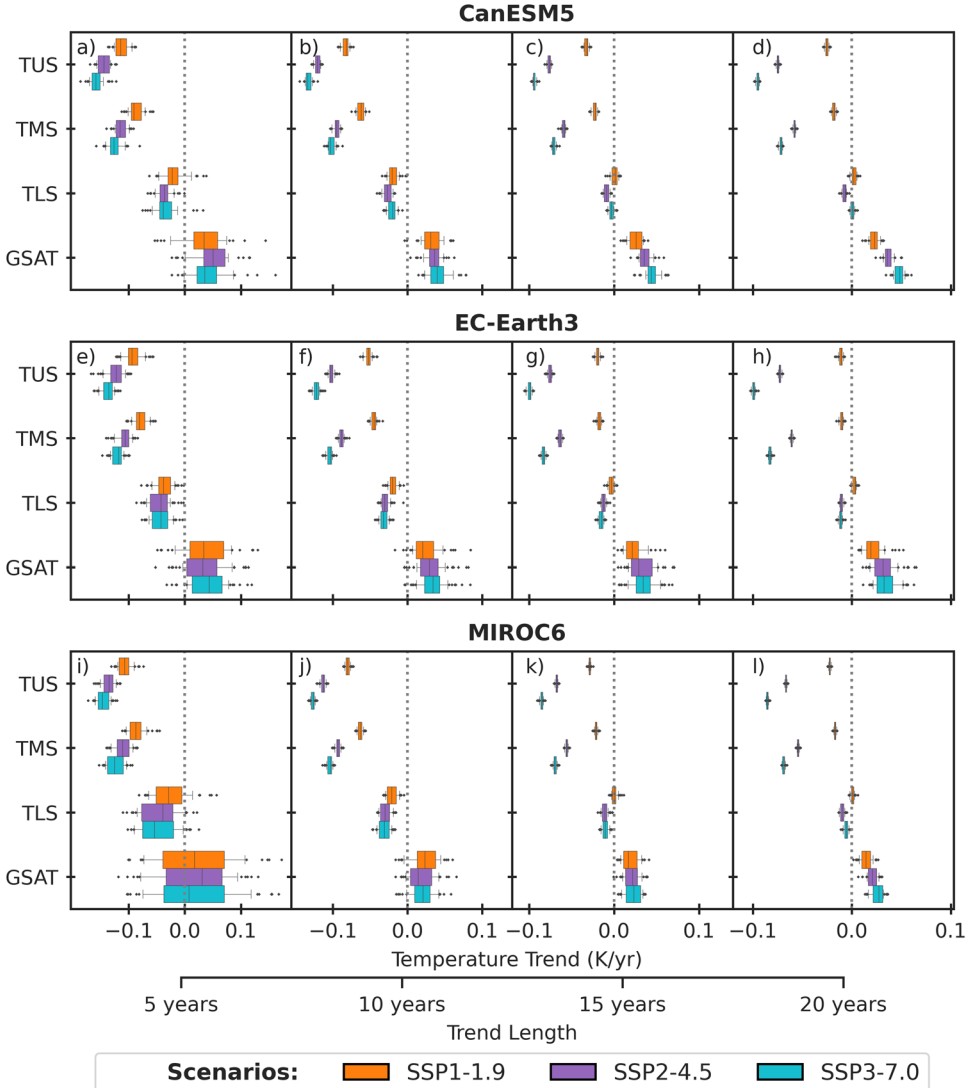

**Fig. 2 | Synthetic observable near-term global annual mean temperature trends [K/year].** Linear trends beginning in 2023 for (**a**, **e**, **i**) 5, (**b**, **f**, **j**) 10, (**c**, **g**, **k**) 15, and (**d**, **h**, **l**) 20-year periods. Data are for synthetic temperatures for temperature of the upper stratosphere (TUS), temperature of the middle stratosphere (TMS), temperature of the lower stratosphere (TLS), and global surface air temperature (GSAT) for three climate models (rows). Boxes show the 25–75th percentile range, whiskers show the 10–90th percentile range, and dots show outliers. Source data are provided as a Source Data file.

year period. This is because the TLS measurement captures both tropical upper tropospheric warming and extratropical lower stratospheric conditions, leading to cancellation across latitudes and relatively similar global mean trends amongst the scenarios in the next 5–10 years.

In contrast, for all trend lengths considered, the ensemble spread in middle and upper stratospheric temperature trends (TMS and TUS) is smaller than for GSAT and TLS. The distributions of TMS and TUS trends between emissions scenarios begin to separate within 5–10 years. The scenarios diverge more rapidly for TUS because $CO_2$-driven cooling of the stratosphere increases with altitude[8]. As expected, the separation of TMS and TUS trends is largest for the most strongly contrasting forcing scenarios (i.e., SSP1–1.9 vs. SSP3–7.0). However, the trends in more similar forcing scenarios (i.e., SSP1–1.9 and SSP2–4.5) also separate within 5–10 years, even after accounting for internal variability.

To further examine the statistical differences in temperature trends between the emissions scenarios, we quantify the overlap of the trend distributions between each combination of scenarios for the different trend lengths. For the TMS and TLS layers (Fig. 3a, b), the overlap in temperature trends between all combinations of scenarios diminishes rapidly to near-zero overlap within 5–10 years, in stark contrast to GSAT and TLS trends (Fig. 3c, d). If we were to follow SSP1–1.9 and use SSP3–7.0 as the counterfactual path for a future with weak mitigation, the reduction in TUS cooling would be clearly detectable in 5 years (blue lines Fig. 3a). If we were to follow SSP1–1.9 and instead used SSP2–4.5 as the counterfactual pathway in which the majority of current mitigation commitments are implemented, there is more overlap of trends over a 5-year window, reaching up to ~20% in one model, but this diminishes rapidly to near-zero for a 10-year trend, indicating the weakening cooling trend could be robustly attributed to the effects of mitigation on this timeframe. The same level of statistical confidence for GSAT trends comparing SSP1–1.9 and SSP2–4.5 is not achieved until after at least 20 years in CanESM5 or longer in EC-Earth3 and MIROC6, which have lower values of TCR and ECS (see Methods).

## Discussion
Our results show that rapid and deep greenhouse gas emission cuts would affect the climate system in a way that could be robustly detected from observed middle and upper stratospheric temperature

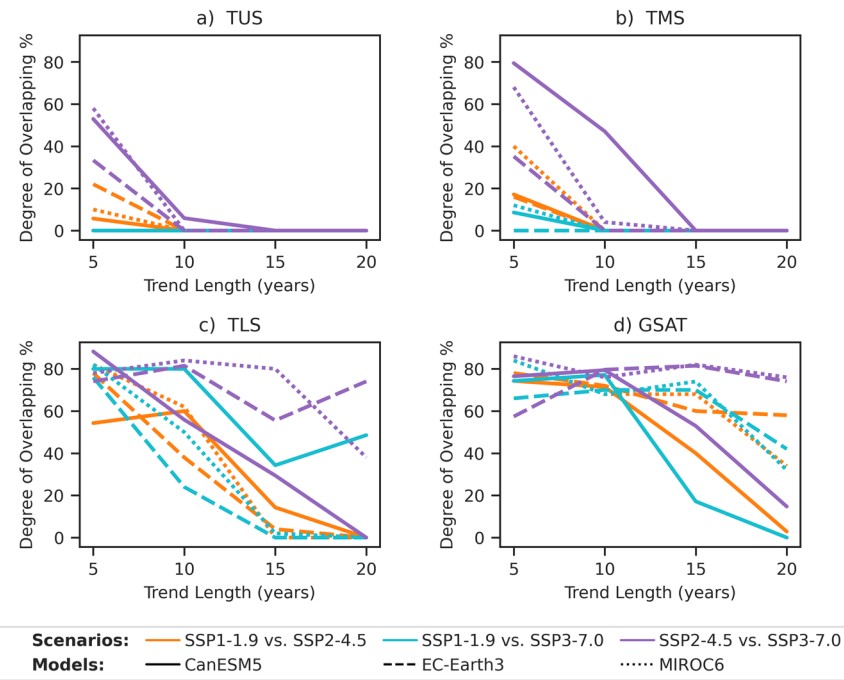

**Fig. 3 | Overlap of global annual mean temperature trend distributions.** Percentage overlap of temperature trend distributions in two contrasting emission scenarios (colours; see Methods). Data are for **a** temperature of the upper stratosphere (TUS), **b** temperature of the middle stratosphere (TMS), **c** temperature of the lower stratosphere (TLS), and **d** global surface air temperature (GSAT) for three climate models (CanESM5 solid, EC-Earth3 dashed, MIROC6 dotted lines). Source data are provided as a Source Data file.

trends within 5–10 years, as compared to a counterfactual world where emissions approximately follow existing climate commitments. In contrast, the model projections indicate it would take at least 20 years to achieve a similar level of statistical power for global surface temperature trends (see also ref. 3,7). Though the global average surface temperature is necessarily a key climate indicator given the Paris Agreement temperature target and its connection to climate risk and impacts[24], our results motivate a wider survey of the climate system to identify other indicators that possess similar signal-to-noise characteristics to global stratospheric temperature, which could be used as part of a multivariate assessment of the effects of mitigation on the climate system. Such evidence would be an important motivation for governments, policymakers and society that their actions are having observable impacts on the climate system and should be sustained in the long-term[25].

## Methods

### Climate models

We use output from three models from the Sixth Coupled Model Intercomparison Project (CMIP6;[26]) that provide large initial condition ensembles (≥30 members) for three ScenarioMIP future emission scenarios: SSP1–1.9, SSP2–4.5, and SSP3–7.0[22]. The models are CanESM5, EC-Earth3, and MIROC6 (see Table 1). These models have different effective climate sensitivities (ECS) and transient climate responses (TCR) that represent higher and lower warming models from the CMIP6 dataset (CanESM5: ECS = 5.64 °C, TCR = 2.66 °C; EC-Earth3: ECS = 4.10 °C, TCR = 2.38 °C; MIROC6: ECS = 2.60 °C, TCR = 1.52 °C;[23,27]).

While the Earth System Grid Federation (ESGF) provides 50 members from CanESM5 (25 initial condition perturbations, r1–25, for two model versions with slightly different physics, p1–2), some of the members were found to exhibit outlier spikes in global annual mean temperatures in individual years that are evident at all atmospheric levels; this suggests a problem with the data quality in those realisations. To remove these spurious data from the analysis, we calculated

### Table 1 | Ensemble members used for each model

| Model | Scenario | No. members | Ensemble IDs |
|---|---|---|---|
| CanESM5 | SSP1–1.9 | 35 | r[1, 4–6, 10–12, 14, 16, 18–20, 22–25]i1p1f1 |
| | | | r[3–7, 9–16, 19–22, 24–25]i1p2f1 |
| | SSP2–4.5 | 34 | r[3–4, 6–9, 12–14, 16–17, 20, 22–23]i1p1f1 |
| | | | r[1–3, 5, 7–8, 10–17, 20–25]i1p2f1 |
| | SSP3–7.0 | 36 | r[1, 3, 5, 7, 9–18, 21–22, 24]i1p1f1 |
| | | | r[1–5, 7, 9–10, 12–16, 18–19, 21–22, 24–25]i1p2f1 |
| EC-Earth3 | SSP1–1.9 | 50 | r[4, 101–149]i1p1f1 |
| | SSP2–4.5 | 54 | r[1, 10, 11, 14–19, 101–119, 128–135, 140–142, 144–150]i1p1f1 |
| | | | r[1, 10, 12, 13, 16–19]i1p1f2 |
| | SSP3–7.0 | 57 | r[1, 4–6, 11, 13, 15, 101–150]i1p1f1 |
| MIROC6 | SSP1–1.9 | 50 | r[1–50]i1p1f1 |
| | SSP2–4.5 | 50 | r[1–50]i1p1f1 |
| | SSP3–7.0 | 50 | r[1–50]i1p1f1 |

the inter-ensemble standard deviation in each year and averaged it over the 20-year analysis period, 2023 to 2042. We then excluded members with temperature anomalies exceeding ±3σ from the ensemble mean in any year during 2023–2042; this filtering removed between 14 and 16 members from each future emissions scenario considered. The remaining members used in the analysis are listed in Table 1.

### Greenhouse gas emissions scenarios

The analysis uses three shared socioeconomic pathway (SSP) scenarios. SSP1–1.9 is in broad alignment with the Paris Agreement 1.5 °C target;[6] assessed there is a >50% likelihood that under SSP1–1.9, GSAT

will remain below 1.6 °C throughout the 21st century. Implementation of global Intended Nationally Determined Contributions (INDCs), which were communicated to the UNFCCC as of 4 April 2016, would achieve global greenhouse gas emissions by 2030 that is similar to scenario SSP2–4.5 (see Fig. 8 of ref. 28); the Nationally Determined Contributions (NDCs) within the registry as of 25 September 2023 put 2030 greenhouse gas emissions ~3–9 GtCO$_2$eq/year (5–15%) below SSP2–4.5 levels (see Fig. 8 of ref. 28). Finally, SSP3–7.0 represents a 'baseline' scenario without near-term mitigation and a continued increase in global greenhouse gas emissions by 2030[22]. It is important to note there remain substantial unknowns about future climate policy and the scenarios do not represent predictions of future greenhouse gas emissions. Furthermore, there are many other emission scenarios that were analysed by ref. 29, but which have not been simulated in comprehensive global climate models; consequently, the three SSPs used here were partly motivated by the availability of suitable climate model simulations.

The ScenarioMIP simulations are initialised on 1 January 2015, with the ensemble members using initial conditions taken from the CMIP6 DECK 'historical' experiment. From this date onwards, the three SSP scenarios follow different evolutions of anthropogenic external forcings (greenhouse gases, aerosols, land use change) and also include natural forcings (solar irradiance and volcanic forcing). In Fig. 3, we compare temperature trends between the three unique permutations of pairs of the SSP scenarios starting in 2023. By this date, the climate states in the individual realisations have already diverged from their initial conditions, both due to external forcings and internal variability.

### Satellite weighting functions

We use the monthly mean 'ta' variable model output, providing atmospheric temperatures on 17 standard pressure levels up to 1 hPa, and the 'tas' variable providing near-surface air temperature. All data are globally and annually averaged. GSAT is calculated as in ref. 30.

We apply three weighting functions to the climate model output to produce synthetic satellite observable atmospheric layer temperatures. The TLS weighting function is from Remote Sensing Systems (http://www.remss.com/) and peaks near 18 km. The TMS and TUS weighting functions are based on Stratospheric Sounding Unit 1 and 2 channels, peaking near 30 km and 37 km, respectively, with weighting functions taken from NOAA STAR SSU version 3 dataset (https://www.star.nesdis.noaa.gov/smcd/emb/mscat/) following[21,31]. While the actual weighting functions are weakly dependent on latitude (e.g., ref. 21), for simplicity, we use constant weighting functions, which are shown in Fig. S1. The functions were interpolated to the models' 17 fixed pressure layers. Previous work has shown that data up to 1 hPa are sufficient to sample the SSU2 channel[16]. Some studies have used the 'plev39' variable from the AerChemMIP project to give higher vertical resolution modelled temperatures. This variable was not available for all the models and scenarios we analysed in this study. However, we tested the sensitivity of the TUS trends to the inclusion of levels above 1 hPa using CanESM5, which provided the 'plev39' variable for its large ensemble for the SSP3–7.0 scenario. This showed that including the 10 additional layers at pressures below 1 hPa has virtually no effect on the calculated near-term TUS trends (see Fig. S2).

### Statistical methods

For each ensemble member, we calculate least squares linear trends of length 5, 10, 15, and 20 years with a start year of 2023. We quantify the difference in trends between each pair of scenarios using the overlap of percentiles. For each comparison, we started by focusing on the higher forcing scenario, i.e., SSP2–4.5 in the SSP1–1.9 vs. SSP2–4.5 comparison, calculating the 90th percentile and 10th percentile trend values. These percentile limits were chosen to ensure that the results would not be strongly influenced by outlier points. Next, we calculate which percentiles of the lower forcing scenario, i.e., SSP1–1.9 in the SSP1–1.9 vs. SSP2–4.5 comparison, these values correspond to. The degree of overlap is defined as the difference between the corresponding percentiles of the lower forcing scenario. This measure of overlap is bounded at 100%.

### Data availability

CMIP6 data was downloaded from the Earth System Grid Federation. The TMS (SSU1) and TUS (SSU2) weighting functions were accessed from the NOAA STAR ftp site: https://www.star.nesdis.noaa.gov/pub/smcd/emb/mscat/data/SSU/SSU_v3.0/. The TLS (MSU4) weighting function was accessed from Remote Sensing Systems ftp site: https://data.remss.com/msu/weighting_functions/std_atmosphere_wt_function_chan_tls.txt. Source data are provided with this paper.

### Code availability

The post-processed global average temperature data generated in this study and plotting codes have been deposited in https://github.com/Bezerragrasi/Projected-temperature-trends.

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

## Acknowledgements

The authors acknowledge funding from the EU H2020 CONSTRAIN project and the Leverhulme Trust. We acknowledge the World Climate Research Programme, which, through its Working Group on Coupled Modelling, coordinated and promoted CMIP6. We thank the climate modeling groups for producing and making available their model output, the Earth System Grid Federation (ESGF) for archiving the data and providing access, and the multiple funding agencies that support CMIP6 and ESGF. We are grateful to Kane Stone for alerting us to the issue with some CanESM5 data.

## Author contributions

A.C.M. conceived the study. G.R.B. acquired the data, performed the analysis, and produced the figures. A.C.M. wrote the manuscript with edits from G.R.B.

## Competing interests

The authors declare no competing interests.
