## [Peer Review File · Nature Communications]

Projected rapid response of stratospheric temperature to stringent climate mitigationREVIEWER COMMENTS

Reviewer #1 (Remarks to the Author):

Romanzini-Bezerra & Maycock presents an intriguing analysis of cooling trends influenced by robust greenhouse gas (GHG) mitigation measures. This study is particularly noteworthy as it provides a method to rapidly assess the impact of mitigation efforts within a 5 to 10-year timeframe, offering timely and relevant scientific evidence. Such an approach, distinct from long-term climate goals, is of great value to both the scientific community and policymakers, addressing the critical question: "Are our actions effective?"

Despite the manuscript's compelling content and potential impact, I regret to inform that I cannot recommend its publication in Nature Communications. The manuscript requires substantial improvements in depth and clarity. here are specific comments:

1) Article Type Clarification: It is unclear whether this submission is intended as a full-length research article or a short communication. While it is categorized as an "article" in the submission system, it appears significantly shorter in comparison to others, especially in the 'Results' section, which comprises only four paragraphs and three figures. Furthermore, the concluding section lacks the depth typically expected in a full-length article. Clarification and possible expansion are necessary to meet the publication standards.

2) Structural Improvements: The manuscript currently lacks a clear structure, making it challenging to distinguish between the introduction and the results sections. Incorporating a few concise section titles would greatly enhance clarity and readability.

3) Methodological Detailing: The Section 2.2, is overly simplified and lacks clarity. The phrase "following earlier work [16]" is too vague for readers. Given the brevity of this section, there is ample opportunity to provide more detailed explanations. Moreover, the relevance of the chosen RCP-SSPs in answering the research question "Are our actions working?" is questionable, considering these are diagnostic scenarios. A discussion on more policy-driven scenarios, such as NDC or net-zero pathways, would be more appropriate.

4) Abstract Formatting: While a minor point, the abstract does not adhere to the specific guidelines recommended for Nature-family journals. It comes across as too simplified and would benefit from a more detailed and structured approach.

Reviewer #2 (Remarks to the Author):

Rapidly observable temperature signals following stringent climate mitigation

G. Romanzini-Bezerra & A. Maycock

Summary

This manuscript analyzes the sensitivity of model-simulated temperature trends to different climate scenarios over the next two decades. Although trends at all levels (surface and lower and mid-stratosphere) show sensitivity to emissions scenario, the difference in mid-stratospheric cooling across scenarios is most pronounced. Using climate model large ensemble experiments, the authors show that mid-stratospheric cooling has little overlap between emissions scenarios on timescales longer than ten years. On the other hand, surface warming and lower stratospheric cooling are less distinguishable across scenarios (i.e., surface warming and lower stratospheric cooling are either similar or nearly the same, depending on the model and experiments considered). This suggests that stratospheric temperature trends would likely provide timely evidence for changes in greenhouse gas emissions.

This manuscript is well-written, has no substantial technical issues, and could plausibly constitute “an important advance to specialists in the field.” This manuscript is quite brief; as far as I can tell, this manuscript is not a content type that would severely limit the word / figure count. If that is correct, it would be useful to cite more relevant literature (on detection and attribution with atmospheric temperature and studies that make use of SSU data), some of which overlaps with this research. Given the aims and scope of the journal it would be helpful to further motivate this work (if possible; see major comment).

Major Comment

Framing: One question I have when I see detection work of this nature (including also a recent Santer et al., 2023 study) is whether detectable global average changes in stratospheric temperature are meaningful for society and/or policy. For example, the abstract notes that there will be “demands for evidence that our actions are benefiting the climate.” Is a reduction in stratospheric cooling a benefit to the climate on its own? Does stratospheric temperature offer benefits relative to atmospheric CO₂ concentration? Or top-of-atmosphere energy imbalance? Perhaps it has better signal properties or is uniquely interesting (because of measurement accuracy or because it is an independent measure of climate policy effectiveness)? If possible, it would be helpful to further motivate stratospheric temperature as an index.

Specific Comments

Please add line numbers in subsequent submissions.

Abstract: Consider clarifying by changing “towards” to “in order to meet”

Abstract: The first statement assumes that the world is going to make deep cuts in carbon emissions. If you have enough words, it would be helpful to break this into two thoughts/sentences: 1) cuts are needed in order to meet Paris targets and 2) such emissions reductions would likely be met with a demand for evidence of effectiveness.

Main Text, first sentence: I am surprised that IPCC WG1 determined we are not on track for a 1.5C target – is this the correct citation?

Main Text, last sentence of paragraph 2: Wouldn't tropospheric temperature variability exceed stratospheric temperature variability in piControl simulations? It's not clear that it is radiative balance that makes the stratosphere less variable.

Main Text, last sentence, paragraph 3: Would "analogous" be more accurate than "equivalent"?

Re: SSP3-7.0: It seems like SSP3-7.0 is plausible, but note that it is unique (doi: 10.1038/s41558-023-01883-2).

Footnote 1: In this case, why not just label this as an AMSU (or ATMS) channel? Or label it as mid-stratosphere (with the text making clear what instrument measurement is equivalent to)? I don't have strong feelings about this, but this label might not be as relevant in this instance?

I'm not sure what manuscript rules apply to this submission, but it could be worth showing the time series (and spread) for different SSP experiments (if length allows). This would help the reader visualize the time-dependent separation across SSP experiments. Even if you only did this for one example model (in order to decrease the number of panels), this could be helpful context.

Figure 2: I think you did this calculation in 5-year intervals. Could this calculation be performed as a function of timescale (5, 6, 7, ... years) to increase the temporal resolution?

Methods: Consider also noting the models ECS values (which might be more familiar to readers; Zelinka et al., 2020: doi: 10.1029/2019GL085782).

Methods: Could you cite/note where the weighting functions come from?

Note that a recent Santer et al (2023; doi: 10.1073/pnas.2300758120) study has similar conclusions (rapid detection of a human fingerprint when incorporating SSU data).

Re: 1hPa model top. What fraction of the SSU2 weighting function is above 1hPa? It seems like the Mitchell et al. study decided on this top somewhat informally. Could you test the sensitivity of SSU2 trends to data availability above 1hPa using data that has a higher top (e.g., from AerChemMIP)? Thompson et al. (doi: 10.1038/nature11579) seemed to use a more conservative cutoff.

Reviewer #3 (Remarks to the Author):

Review of “Rapidly observable temperature signals following stringent climate mitigation” by Romanzini-Bezerra and Maycock

The paper is a nicely written and argued short form submission. The result is new and novel, although not particularly surprising from a basic physics perspective. It has high policy relevance and will be of interest to a generalist audience. I would therefore recommend publication following some minor revisions for clarity and following consideration of one major comment relating to a possible addition of a discussion of generalising the approach to create a multivariate indicator metric in future.

Major comments

1. The implication of your concluding paragraph is that the stratospheric temperatures would uniquely provide an indicator of successful mitigation. I'm not sure this is the case. Rather they highlight the characteristics of the type of indicator we might try to find and, indeed, any framework might be more robust were it multivariate. I could foresee other integrative measures such as global OHC or others potentially also having preferable characteristics. It would surely be worth a very brief paragraph highlighting how finding a suite of such indicators across the climate system would increase confidence in making a conclusion that mitigation is having effect?

Minor comments

1. In the abstract you talk of Paris Agreement temperature targets whereas there is one singular temperature target as detailed in article 2a of the agreement text. I would modify the text to speak to the Paris Agreement temperature target accordingly.

2. It feels a little odd to label the two stratospheric channels SSU1 and SSU2 given that SSU ceased operation some decades ago and the continuation of these series relies upon measurements from newer platforms (even beyond AMSU-A to ATMS etc.) not termed SSU as you correctly note in the footnote. Maybe for readability terming these TMS and TUS or similar would be clearer as like TLS it would at least refer to the approximate portion of the stratosphere being sampled.

3. The statement in page 2 regarding the observed trajectory being equivalent to a single member of the ensemble has quite a lot of hefty assumptions underlying it and lacks a supporting reference. I'm not sure its needed so one option is to delete it. Another would be to make clearer some of these assumptions or at least add a supporting reference.

4. I think you need to clarify what you mean by pairs of scenarios in the top of page 3. Are these simulations started from identical initial conditions but run with distinct forcings moving forwards? This may be able to be handled via edits to 2.1 to make this clear.

Reviewer #1

Romanzini-Bezerra & Maycock presents an intriguing analysis of cooling trends influenced by robust greenhouse gas (GHG) mitigation measures. This study is particularly noteworthy as it provides a method to rapidly assess the impact of mitigation efforts within a 5 to 10-year timeframe, offering timely and relevant scientific evidence. Such an approach, distinct from long-term climate goals, is of great value to both the scientific community and policymakers, addressing the critical question: "Are our actions effective?"

Despite the manuscript's compelling content and potential impact, I regret to inform that I cannot recommend its publication in Nature Communications. The manuscript requires substantial improvements in depth and clarity.

1) Article Type Clarification: It is unclear whether this submission is intended as a full-length research article or a short communication. While it is categorized as an "article" in the submission system, it appears significantly shorter in comparison to others, especially in the 'Results' section, which comprises only four paragraphs and three figures.

The manuscript was initially submitted to another Nature journal as a 'Brief Correspondence' which has strict length limits. The manuscript was later transferred to Nature Communications. Nature Communications does not publish 'Brief Correspondence' pieces and hence the manuscript was registered as an Article. We were not asked to make any edits to the manuscript before it was sent for review and there is no minimum Article length in Nature Communications. We think the article works well in a succinct format and that the key conclusions of our work can be communicated with relatively short text and figures. However, in light of the reviewer's comments and the switch in manuscript type giving more flexibility in length, we have taken on board their points and extended the manuscript to include more detail as outlined below.

Furthermore, the concluding section lacks the depth typically expected in a full-length article. Clarification and possible expansion are necessary to meet the publication standards.

We have extended the Discussion section and added text to address the comment of Reviewer 3 on the potential for multivariate detection of mitigation effects, which could include stratospheric temperatures alongside other climate variables with similar signal-to-noise properties. We have also been more specific in the Discussion about the specific results of the study including the timescales for emergence of differences in trends between SSP forcing scenarios to achieve similar signal-to-noise levels: for SSP1-1.9 vs. SSP3-7.0, a 5 year TUS trend gives comparable statistical properties to a ~20 year trend for GSAT.

2) Structural Improvements: The manuscript currently lacks a clear structure, making it challenging to distinguish between the introduction and the results sections. Incorporating a few concise section titles would greatly enhance clarity and readability.

We have added section headings following the Nature Communications guidelines to create a clearer structure to the Article including Abstract, Introduction, Results and Discussion.

3) Methodological Detailing: The Section 2.2, is overly simplified and lacks clarity. The phrase "following earlier work [16]" is too vague for readers. Given the brevity of this section, there is ample opportunity to provide more detailed explanations.

We have removed 'following earlier work' and have expanded the Methods section including the part about Satellite Weighting Functions as follows:

"We apply three satellite weighting functions to the climate model output to produce synthetic observable temperature metrics. The TLS weighting function is from Remote Sensing Systems (<http://www.remss.com/>) and peaks near 18km. The TMS and TUS weighting functions are based on Stratospheric Sounding Unit 1 and 2 channels, peaking near 30km and 37km respectively, with weighting functions taken from NOAA STAR SSU version 3 dataset (<https://www.star.nesdis.noaa.gov/smcd/emb/mscat/>) following (Zou et al 2014; Zou and Qian 2016). While the actual weighting functions are weakly dependent on latitude (e.g., Zou and Qian 2016), for simplicity we use constant weighting functions, which are shown in Figure S1. The functions were interpolated to the models' 17 fixed pressure layers. Some studies have used the 'plev39' variable from the AerChemMIP project to give higher vertical resolution of modelled atmospheric temperatures (e.g., Thompson et al., 2012; Santer et al., 2023). This may be particularly important for the TUS channel which peaks near 37km and has part of the weighting function at pressures less than 1hPa (Figure S1). This variable was not available for all the large ensemble simulations and scenarios we analyse in this study. However, we tested the sensitivity of the TUS trends to the inclusion of levels above 1hPa using CanESM5, which provided the 'plev39' variable for its large ensemble for the SSP3-7.0 scenario. This showed that including the 10 additional layers at pressures below 1hPa has virtually no effect on the calculated near-term TUS trends (see Figure S2). Given the limited vertical extent of the model data, we do not include the SSU3 channel in the analysis, which peaks near 43km and covers the upper stratosphere/lower mesosphere (e.g., Zou and Qian, 2016)."

Moreover, the relevance of the chosen RCP-SSPs in answering the research question "Are our actions working?" is questionable, considering these are diagnostic scenarios. A discussion on more policy-driven scenarios, such as NDC or net-zero pathways, would be more appropriate.

While we agree there are a large number of NDC and net zero scenarios available e.g., IPCC AR6 WGIII (2022), our analysis is limited in scenario selection by the choices of CMIP6 ScenarioMIP, who produced simulations for selected SSPs (Boer et al., 2016). The 5 illustrative scenarios ScenarioMIP prioritised (SSP1-1.9, SSP1-2.6, SSP2-4.5, SSP3-7.0, SSP5-8.5) formed the basis of the projections assessed in IPCC AR6 WGI Chapter 4 (Lee et al., 2021). Rather than including all the ScenarioMIP SSPs in our study, we chose to omit some. We did not include SSP5-8.5, since this has been criticised as being unrealistic based on current policies (Hausfather and Peters, 2020). We selected SSP1-1.9 as the scenario most consistent with the Paris temperature goal because Lee et al. (2021) assessed "It is *more likely than not* that under SSP1-1.9, GSAT relative to 1850–1900 will remain below 1.6°C throughout the 21st century, implying a potential temporary overshoot above 1.5°C of no more than 0.1°C'. While there aren't ScenarioMIP simulations available based on current NDCs, the current NDCs as of 2023 place us just below SSP2-4.5 in terms of global greenhouse gas emissions at 2030 (see Figure 8 of UNFCCC, 2023). Therefore, while

idealised, the scenarios compared in the manuscript do bear some relation to current policies and the Paris target. We have expanded the selection and description of the scenarios in the Methods.

References

Hausfather and Peters, *Nature* 577, 618-620 (2020) doi: <https://doi.org/10.1038/d41586-020-00177-3>

UNFCCC, 2023, Nationally determined contributions under the Paris Agreement. Synthesis Report by the United Nations Framework Convention on Climate Change. Conference of the Parties serving as the meeting of the Parties to the Paris Agreement Fifth session. 14 November 2023. FCCC/PA/CMA/2023/12.

Lee, J.-Y. et al. Future Global Climate: Scenario-Based Projections and Near-Term Information. In *Climate Change 2021: The Physical Science Basis. Contribution of Working Group I to the Sixth Assessment Report of the Intergovernmental Panel on Climate Change.* (eds Masson-Delmotte, V., P. Zhai, A. Pirani, et al.). (Cambridge University Press, Cambridge, United Kingdom and New York, NY, USA) pp. 553–672. <https://doi.org/10.1017/9781009157896.006> (2021)

4) Abstract Formatting: While a minor point, the abstract does not adhere to the specific guidelines recommended for Nature-family journals. It comes across as too simplified and would benefit from a more detailed and structured approach.

The Brief Correspondence this was originally formatted for had a 100 word limit for the abstract. We have amended the abstract to better fit Nature Communications Article guidelines (<200 words).

Reviewer #2

Summary

This manuscript analyzes the sensitivity of model-simulated temperature trends to different climate scenarios over the next two decades. Although trends at all levels (surface and lower and mid-stratosphere) show sensitivity to emissions scenario, the difference in mid-stratospheric cooling across scenarios is most pronounced. Using climate model large ensemble experiments, the authors show that mid-stratospheric cooling has little overlap between emissions scenarios on timescales longer than ten years. On the other hand, surface warming and lower stratospheric cooling are less distinguishable across scenarios (i.e., surface warming and lower stratospheric cooling are either similar or nearly the same, depending on the model and experiments considered). This suggests that stratospheric temperature trends would likely provide timely evidence for changes in greenhouse gas emissions.

This manuscript is well-written, has no substantial technical issues, and could plausibly constitute “an important advance to specialists in the field.” This manuscript is quite brief; as far as I can tell, this manuscript is not a content type that would severely limit the word / figure count. If that is correct, it would be useful to cite more relevant literature (on detection and attribution with atmospheric temperature and studies that make use of SSU data), some of which overlaps with this research. Given the aims and scope of the journal it would be helpful to further motivate this work (if possible; see major comment).

We thank the reviewer for their positive comments about our study and their constructive suggestions for improvements. We have taken on board their feedback, in particular to include more discussion of previous literature related to attribution of temperature trends, and to provide more context to motivate climate monitoring.

Major Comment

Framing: One question I have when I see detection work of this nature (including also a recent Santer et al., 2023 study) is whether detectable global average changes in stratospheric temperature are meaningful for society and/or policy. For example, the abstract notes that there will be “demands for evidence that our actions are benefiting the climate.” Is a reduction in stratospheric cooling a benefit to the climate on its own? Does stratospheric temperature offer benefits relative to atmospheric CO₂ concentration? Or top-of-atmosphere energy imbalance? Perhaps it has better signal properties or is uniquely interesting (because of measurement accuracy or because it is an independent measure of climate policy effectiveness)? If possible, it would be helpful to further motivate stratospheric temperature as an index.

Thanks for raising this important point. We agree that stratospheric temperatures are not directly connected to climate change impacts in the way of global average surface temperature. Nevertheless, we believe objectively observable climate indicators can still constitute a valuable message to governments and wider society that mitigation is affecting the climate, even remotely from surface. This is preferable to saying that we need to wait 20+ years to be sure that actions are having an effect. It was noted as far back as the IPCC FAR (1990) that “Stratospheric cooling alone has been suggested as an important detection

variable". We have also added to the Introduction to describe these early links of stratospheric temperatures to IPCC. Attribution of atmospheric temperature trends remains a key part of the IPCC WGI Assessment today (see IPCC AR6 WGI Chapter 3). On the suggestion of Reviewer 3, we have also amended the Discussion to no longer portray that stratospheric temperatures are the only climatic variable that possess good signal-to-noise characteristics and have added a sentence there stating the need to seek multivariate signals for the effects of mitigation on the trajectory of climate.

Specific Comments

Please add line numbers in subsequent submissions.

Noted and sorry this was missed.

Abstract: Consider clarifying by changing "towards" to "in order to meet"

Changed.

Abstract: The first statement assumes that the world is going to make deep cuts in carbon emissions. If you have enough words, it would be helpful to break this into two thoughts/sentences: 1) cuts are needed in order to meet Paris targets and 2) such emissions reductions would likely be met with a demand for evidence of effectiveness.

Thanks for this suggestion. The text has been amended to distinguish these two points.

Main Text, first sentence: I am surprised that IPCC WG1 determined we are not on track for a 1.5C target – is this the correct citation?

Thanks. The citation was meant to be IPCC AR6 WGIII, but we have updated the citation to the UNEP 2023 Emissions Gap Report as this makes an explicit statement on the emissions gap.

Main Text, last sentence of paragraph 2: Wouldn't tropospheric temperature variability exceed stratospheric temperature variability in piControl simulations? It's not clear that it is radiative balance that makes the stratosphere less variable.

Yes, they would still be more variable. The larger tropospheric temperature variability is predominantly driven by internally (self) generated climate variability on interannual to decadal timescales associated with variations in global ocean heat uptake, as well as other related coupled atmosphere-ocean processes such as cloud radiative effects. Conversely, the near global radiative balance of the stratosphere means that internally generated variability is extremely small and only externally-forced changes in heating rates (either from natural or human forcings) induce temperature change. This text has been reworked and expanded.

Main Text, last sentence, paragraph 3: Would "analogous" be more accurate than "equivalent"?

Changed

Re: SSP3-7.0: It seems like SSP3-7.0 is plausible, but note that it is unique (doi: 10.1038/s41558-023-01883-2).

Thanks for this interesting paper. Since we do not aim to evaluate whether any of the scenarios investigated are more or less realistic we have not included a discussion of this.

Footnote 1: In this case, why not just label this as an AMSU (or ATMS) channel? Or label it as mid-stratosphere (with the text making clear what instrument measurement is equivalent to)? I don't have strong feelings about this, but this label might not be as relevant in this instance?

For clarity and consistency and on the suggestion of Reviewer 3, we have relabeled SSU1 and SSU2 to TMS (Temperature Middle Stratosphere) and TUS (Temperature Upper Stratosphere) throughout the manuscript, so they are not specific to a satellite instrument that might be superseded (e.g. SSU or AMSU-A). This footnote has been removed and the height ranges and weighting functions for TMS and TUS are explained in the Methods.

I'm not sure what manuscript rules apply to this submission, but it could be worth showing the time series (and spread) for different SSP experiments (if length allows). This would help the reader visualize the time-dependent separation across SSP experiments. Even if you only did this for one example model (in order to decrease the number of panels), this could be helpful context.

Figure R1 shows the temperature anomaly timeseries spanning 2023 to 2045 for the CanESM5 model. The thick line shows the ensemble mean and the shading shows the 10-90th percentile of the ensemble spread. Anomalies are calculated relative to the period 2015 to 2022. We have added the figure to the main text as new Figure 1.

Figure R1: Timeseries of annual mean temperature anomalies in CanESM5 with respect to the period 2021-25, as used in the linear trend calculations. Thick lines show the ensemble mean and shading denotes the 10-90th percentiles of the ensemble spread.

I think you did this calculation in 5-year intervals. Could this calculation be performed as a function of timescale (5, 6, 7, ... years) to increase the temporal resolution?

Yes, in the manuscript we calculated the degree of overlap using 5-year increments in trend length. We have recalculated this using an annual increment with the results shown Figure R2. At periods <5 years, the overlap in the TUS and TMS trends increases sharply. The higher temporal resolution makes the diagram look busier and harder to interpret, particularly for the TLS and GSAT data where the overlap remains large over a greater range of trend lengths. Consequently, we think the original figure more clearly displays the data so we have not changed it.

Figure R2: As in Figure 2 of the main text but with the resolution increased to annual increments in trend length.

Methods: Consider also noting the models ECS values (which might be more familiar to readers; Zelinka et al., 2020: doi: 10.1029/2019GL085782).

Thanks for the suggestion. We have added the models' ECS values to the Methods.

Methods: Could you cite/note where the weighting functions come from?

We have added the sources of the weighting functions in the Methods and Data Availability sections.

Note that a recent Santer et al (2023; doi: 10.1073/pnas.2300758120) study has similar conclusions (rapid detection of a human fingerprint when incorporating SSU data).

Thanks. We have added a citation to Santer et al (2023) to the Introduction as motivation for our study, describing their result that the detectability of human influence on temperature trends is strongly increased when including stratospheric temperature data.

Re: 1hPa model top. What fraction of the SSU2 weighting function is above 1hPa? It seems like the Mitchell et al. study decided on this top somewhat informally. Could you test the sensitivity of SSU2 trends to data availability above 1hPa using data that has a higher top (e.g., from AerChemMIP)? Thompson et al. (doi: 10.1038/nature11579) seemed to use a more conservative cutoff.

Thanks for raising this important point. Unfortunately, the AerChemMIP plev39 variable was not available for all the large ensemble models and the multiple SSP scenarios required for our analysis. However, we were able to locate this variable for CanESM5 for the SSP3-7.0 scenario. Figure R3 shows the original temperature trend results from the manuscript for the TUS channel compared with the same data with an additional 10 layers added at pressures <1hPa up to 0.03hPa. The addition of these layers makes virtually no difference to the calculated TUS trends. The reason for this is that the peak of the TUS weighting function lies below 1hPa and the temperature for the weighting function is pressure weighted so higher altitudes contribute relatively less than lower layers. We have added Figure R3 to the Supplementary Information.

Figure R3: Comparison of TUS trends in CanESM5 for the original manuscript data (purple) and adding an additional 10 layers at pressures <1hPa up to 0.03hPa (orange). Note we were only able to locate the plev29 data for CanESM5 for the SSP3-7.0 scenario and we were therefore unable to use this for the main analysis in the manuscript.

Reviewer #3

The paper is a nicely written and argued short form submission. The result is new and novel, although not particularly surprising from a basic physics perspective. It has high policy relevance and will be of interest to a generalist audience. I would therefore recommend publication following some minor revisions for clarity and following consideration of one major comment relating to a possible addition of a discussion of generalising the approach to create a multivariate indicator metric in future.

We thank the reviewer for their positive comments about our manuscript. We have considered their suggestions and reply to their points below in blue.

Major comments

1. The implication of your concluding paragraph is that the stratospheric temperatures would uniquely provide an indicator of successful mitigation. I'm not sure this is the case. Rather they highlight the characteristics of the type of indicator we might try to find and, indeed, any framework might be more robust were it multivariate. I could foresee other integrative measures such as global OHC or others potentially also having preferable characteristics. It would surely be worth a very brief paragraph highlighting how finding a suite of such indicators across the climate system would increase confidence in making a conclusion that mitigation is having effect?

Thanks for this important comment. We agree there may be other measures not considered in our work that could provide similar indicators of the effects of mitigation on the climate trajectory. We have added a paragraph in the discussion offering a wider perspective and no longer describe stratospheric temperature trends as a unique indicator.

The final paragraph now reads:

“The results presented here show that stringent climate mitigation in the very near future would divert the trajectory of the physical climate system in a way that could be detected from observations in around 5 years based on examination of middle and upper stratospheric temperature trends. In contrast, it would take 20 years to arrive at this conclusion with a similar level of statistical confidence based on global surface temperature trends (see also Marotzke 2019). Though global average surface temperature will necessarily remain a key measure given the Paris temperature target and the connection to climate risk and impacts, the results in this study motivate a wider survey of the climate system to identify other indicators that possess similar signal-to-noise characteristics to global mean stratospheric temperatures, and could be part of a multivariate assessment of the effects of climate mitigation on the climate system. Such evidence would be an important motivation for governments, policymakers and society that their actions are having observable benefits in the climate system and should be sustained in the long-term (Hegglin et al., 2022).”

Minor comments

1. In the abstract you talk of Paris Agreement temperature targets whereas there is one singular temperature target as detailed in article 2a of the agreement text. I would modify the text to speak to the Paris Agreement temperature target accordingly.

Thanks for spotting this. We have amended the abstract accordingly.

2. It feels a little odd to label the two stratospheric channels SSU1 and SSU2 given that SSU ceased operation some decades ago and the continuation of these series relies upon measurements from newer platforms (even beyond AMSU-A to ATMS etc.) not termed SSU as you correctly note in the footnote. Maybe for readability terming these TMS and TUS or similar would be clearer as like TLS it would at least refer to the approximate portion of the stratosphere being sampled.

Thanks for this suggestion. We have adopted the naming convention TMS and TUS for consistency with TLS and explain the weighting functions and how they relate to current and previous satellite measurements in the Methods.

3. The statement in page 2 regarding the observed trajectory being equivalent to a single member of the ensemble has quite a lot of hefty assumptions underlying it and lacks a supporting reference. I'm not sure its needed so one option is to delete it. Another would be to make clearer some of these assumptions or at least add a supporting reference.

We have amended this text to read: "Due to internal variability, there are many plausible trajectories the climate system could take under the same future boundary conditions and forcing (Deser et al., 2020). Within the large ensemble framework, one can therefore think of the observed climate trajectory as roughly analogous to a single member drawn from the scenario that most closely matches our future emissions path."

4. I think you need to clarify what you mean by pairs of scenarios in the top of page 3. Are these simulations started from identical initial conditions but run with distinct forcings moving forwards? This may be able to be handled via edits to 2.1 to make this clear.

Yes, this is what we mean by pairs of scenarios. The simulations following either SSP1-1.9, SSP2-4.5 or SSP3-7.0 forcings are all initialised on 1 January 2015. We analyse them from current day, starting 2023, so the simulations has already diverged substantially from their initial conditions owing to internal climate variability and the effects of external forcings.

We have added additional details in the Methods section describing the climate models and scenarios in more detail.

REVIEWERS' COMMENTS

Reviewer #2 (Remarks to the Author):

The reviewers have satisfactorily responded to all my comments; I believe this manuscript is suitable for publication.